# Into the “New Normal”: The Ethical and Analytical Challenge Facing Public Health Post-COVID-19

**DOI:** 10.3390/ijerph19148385

**Published:** 2022-07-08

**Authors:** Hagai Boas, Nadav Davidovitch

**Affiliations:** 1Department of Politics and Government, Ben-Gurion University of the Negev and Van Leer Jerusalem Institute, Jerusalem 9214116, Israel; 2Faculty of Health Sciences, School of Public Health, Ben-Gurion University of the Negev, Beer Sheva 8410501, Israel

**Keywords:** new normal, COVID-19, public health ethics, community, bioethics, vaccinations

## Abstract

Even though various countries’ overall policy for dealing with the pandemic was not particularly innovative, the pandemic was perceived as a unique crisis. “COVID exceptionalism” has seemed to create “a new normal” that we all need to “learn to live with”. The main change in perspective, while not new for public health experts, is that health exists within a social and political context. While public health ethics has turned out to be an important discipline, there is a long way to its wider acceptance. Entering the “new normal” calls for a wider embrace of public health approaches to ethics. The renewed emphasis on understanding health as a social concept encompasses central normative implications in relation to dealing with COVID-19 and in relation to dealing with other global crises, chiefly climate change. We argue that entering the era of “the new normal” in healthcare requires a nuanced understanding of the relationship between the individual and society and demands the formulation of a new system of bioethics focused on the concept of solidarity as a central value in public health. Such a concept should refer to the fact that in the “new normal”, risks require new social and political formations of standing together in confronting risks that cross national, cultural, and identity borders. Forming and expanding solidarity in health and healthcare, we argue, is the main normative challenge for public health today.

## 1. Introduction

As of the end of March 2022, coronavirus disease 2019 has led to the death of at least over six million people worldwide, and this is most likely a significant underestimation. The mortality rate in the United States alone is more than one million people, and as of the time of writing of this paper, the entire world has been hit by six waves of the pandemic since its declaration as a public health emergency of international concern and later as a pandemic at the beginning of 2020 [1]. Primarily, the pandemic reinforced public health tenets that health is not only a matter of personal responsibility but is dependent on a complex system of structural contexts—some institutional, others normative, but all emphasizing the public and collective dimensions of health. While we have not yet managed to entirely overcome the pandemic, we already know that beyond considering the virus itself and the burnout of healthcare workers, focusing narrowly on morbidity and mortality rates lead to an incomplete and even harmful response. Thus, damage from the pandemic-induced lockdowns includes weight gain, lack of exercise, increased smoking, loneliness, depression, damage to social capital, poverty, and more [2]. A broad view of public health must also include the social, economic, environmental, and political factors influencing healthcare. In short, the pandemic proved again that health is a social, political, and normative matter, no less than a personal one [3,4].

The crisis highlighted the importance of public health, a discipline that has been pushed too many times to the sidelines of the healthcare system in favor of providing clinical care to patients rather than addressing upstream forces, social determinants of health, and the population’s health behavior. The rapid recruitment of public health as a discipline and public health experts brought on by COVID-19 mobilized different radical interventions but also created an ongoing crisis of confidence between the public and public health experts and leaders. A significant change in the image of public health during the pandemic has been the politicization of public health policies. Although issues of trust and compliance have always accompanied public health policies, the uncertainties, failures, and helplessness that accompanied the pandemic in its first stages damaged the professional reputation of public health as impartial. Instead, throughout the world, public health was conceived as political, leading to large controversies regarding its policies and interventions. Moreover, even though various countries’ overall policy for dealing with the pandemic was not particularly innovative and included traditional measures such as isolation, social distancing, lockdowns, and, from a certain stage, vaccines, these measures were considered contentious, and non-compliance crossed political camps and social identities [5,6]. In contrast to the neo-liberal trends of privatizing and decentralizing health services, public health apparatuses emerged during the pandemic as features of a “strong state” whose “visible hand” now intervenes, controls, and manages the very heart of what just before the pandemic was considered the sacred core of free liberal life [7]. These conflicting trends created a unique position for public health: it became unprecedently present in public life, media, and politics and, as such, became highly politicized to the degree that trust in experts and policymakers has been eroded.

The pandemic was and still is perceived as a unique crisis. “COVID exceptionalism” has seemed to create “a new normal” that we all need to “learn to live with”. Although the intervention of public health introduced no innovative measurements and in fact recommended traditional steps such as quarantines, physical distancing, and mask wearing, the scope of the pandemic and the measures taken to combat it were unprecedented to the degree that the wording “the new normal” was used to describe it [5]. We understand the “new normal” as referring to life at a constant risk which can only be contained and regulated but cannot be totally overcome. The “new normal” can be defined as living in a constant state of “preparedness” [8]. The risks that constitute the “new normal” are global, transgressing spatial boundaries and producing states of uncertainty and instabilities, and pose existential threats. While these characteristics emerged and were discussed in relation to climate change, emerging infectious diseases, and other threats, the pandemic was certainly perceived as one of the shifting points towards “the new normal” [9]. As one of the editorial board members of *The New England Journal of Medicine* asserted: “the COVID-19 pandemic is going to be one of those dichotomous events that divides life into before and after. We live through them, learn from them, and adjust” [10]. The “new normal” entails a more central role for public health in the building of preparedness, but more importantly in the promise to promote better integration of “health in all policies” perspectives within and outside traditional public health spaces. The premise of public health that personal health derives also from collective health, although not new, is essential for living among global health hazards, especially when there is a need for non-traditional public health professionals to be involved, on global and national as well as local scales. In fact, this requires a change in perspective, which, while not new, is yet to be sufficiently implemented. This perspective emphasizes the social and political contexts in which public health dwells. As health also plays out on the social level, it is necessary to apply tools and concepts from the field of social sciences in addressing it [11].

The renewed emphasis on understanding health and healthcare as social concepts encompasses central normative implications in relation to dealing with COVID-19 and other pandemics and in relation to dealing with other global crises, chiefly climate change [12]. We argue that thinking about the sociology of health requires a nuanced understanding of the relationship between the individual and society and requires emphasizing the collective aspects of public health ethics. After more than two decades of institutionalizing public health ethics, it should become much more central and adapted to the “new normal”. We suggest that such an ethics system would be sensitive to the shift from public to publics and to different forms of solidarity that would help confront the risks of the “new normal”.

## 2. Toward a New Bioethics

For many decades, and even today after years of criticism, the link between society and healthcare has been perceived mostly as centered on individuals’ health, the doctor–patient relationship, and the rights and duties this relationship entails. A central foundation of this concept is the patient rights approach, which has found legislative expression in many countries around the world [13]. This approach has defined what is permissible between doctors and patients and what is not. In addition, it has strengthened the patient’s status and autonomy in determining the course of their treatment and outlined the normative boundaries of the relationship between patients and the medical institution [14]. This version of bioethics can be understood as “liberal bioethics”, as it follows the liberal political tradition of focusing on the individual and their will, understanding, decisions, and rights [15]. The liberal foundation of this approach can also be seen in its replication of the concept of negative liberty, i.e., the individual’s freedom from the coercive power of external entities—in this case, the medical institution—and great effort is dedicated to creating a protective space for maintaining the individual’s privacy and rights [16].

More collective dimensions such as “relational autonomy” do exist in bioethics [17]. Justice is one of the four Georgetown principles, appearing already in the Belmont report, which refers to the societal implications of health resources’ distribution, equity, and equality [18]. Feminists and other critics have suggested adjustments to bioethics that focus on the social and political contexts of medical ethics [19,20]. Nonetheless, the “new normal” in general and the pandemic specifically call for a significant revision in bioethics that will render collective values much more central. This was the call of certain interpretations of public health ethics, but unfortunately they were marginalized, at least in the public discourse. As scholars have engaged within the field of public health ethics during the last two decades, we see a current window of opportunities. Under “the new normal”, liberal bioethics is losing currency and a more society-oriented ethics is gaining momentum. Thus, in the age of the pandemic, the patient is no longer only an individual whose rights must be protected, but it is the entire population that must be kept safe. As a result, public health physicians who provide treatment at the population level and who consider the pandemic’s social, economic, and political aspects are now moving into the forefront. The relationship between health and society is now geared toward prevention and containment as well as community resilience on all levels, including the psychological and social levels, rather than being focused on medical treatment alone [21].

Liberal bioethics emerged in the second half of the 20th century in an entirely different context than that from which public health medicine emerged. The latter arose in the 19th century as a direct result of its interaction with social sciences and the rise of the nation state; the 20th century’s welfare state also contributed to the development of public health as a discipline. The unit of analysis in public health medicine is the population, which is a quintessential 19th century concept, and bioethical values based on individualism and the individual’s rights were at first foreign to the medical field [3].

Under the liberal premise, we tend to confuse bioethics with medical ethics. Issues such as informed consent, autonomy, medical confidentiality, and others are related to the relationship that develops between doctors and patients, and their ethical dimension is based on the professional ethos of medicine tracing back to the Hippocratic Oath. However, by referring to the concept of bioethics in a broader way and in relation to public health, the potential for a system of values and rules stemming from a different position emerges. This system is not necessarily intended to replace the individualistic approach but rather to add another layer to it, one that relates to society as a whole and views the good of the individual as derived from the good of the collective, rather than the other way around.

In fact, the linguistic indication of the term “bioethics” pertains to the norms associated with managing the “bios”, meaning life. In terms of liberal bioethics, the focus has been on normative questions related primarily to two fields. The first is patient autonomy regarding health risks related to medical treatments or participation in clinical trials. The second is determining the social norms relating to the use of advanced medical technology, including issues such as genetic information usage and its implications [4]. Liberal bioethical reasoning is based on applying the logic of analytic philosophy and law to the fields of health and medicine. The result of this line of reasoning is clearly expressed in the Georgetown principles, which emphasize the individual’s autonomy and the need to obtain consent rather than the medical institution’s responsibility to provide beneficial treatment [22].

Liberal bioethics has many virtues, especially considering the grim history of exploitation, coercion, and oppression of patients. To combat historical transgressions against patient autonomy, a worldview that emphasizes the individual as an actor whose autonomy is sacred became hegemonic. Critics of bioethics argued that in many cases, the social conditions in which this actor operated were moved to the background of the discussions, if they were discussed at all [19,20]. Under the mainstream paradigm of liberal bioethics, gender, class, migration or other crucial social categories played minimal role—proper bioethical management was meant to focus on ensuring their autonomy, abstracted as it may be. The principle of autonomy and the use of informed consent implied that regardless of any social positioning, each human has the fundamental right of self-determination. This axiomatic principle was seen as an effective remedy against the wrongs of exploitation and coercion and became the pillar of liberal bioethics [23].

Nonetheless, this notion of autonomy draws criticism even within the individualistic approach in bioethics; some critics point to the impossibility of attaining fully informed consent, while others challenge the very concept of autonomy [24]. Another line of criticism emphasizes the social conditions in which both healthcare workers and patients are operating. Ultimately, while informed consent and autonomy are guiding principles for bioethics committees in many healthcare institutions, much of the criticism levelled against these principles puts their very feasibility into question [25,26,27].

Unlike liberal bioethics, public health ethics emphasizes the social, economic, and political contexts of healthcare and the system of social and political forces shaping the social decision-making process [28,29]. The gaps between various populations in terms of access to medical care and social and environmental determinants of health such as sanitation, water, and electricity are central to the methodology of public health ethics [30]. Indeed, while both bioethics and public health law help healthcare professionals to identify and respond to legal and moral dilemmas in their work, during the last decade, there has been a growing interest in rethinking public health law and ethics as distinct fields. This perspective has been important to encourage the introduction of social and ethical sensitivities to address public health challenges already long before the COVID-19 pandemic, ranging from reducing health inequities (both within and between countries) to the implementation of the new International Health Regulations (IHRs) almost two decades ago in the context of emerging infectious diseases and pandemic influenza. While traditional bioethical thinking will ask questions regarding scarce resources or the appropriateness of certain measures such as quarantine, vaccination, or animal culling, a public health ethics approach will start from the political power relations and how various communities should react and interact [31].

Indeed, public health ethics is not new and had a strong voice in medicine and bioethics long before the pandemic. The point we want to make here is that although present within circles of professional discussions and debates, the current pandemic introduced these debates outside those circles and into the public sphere of media and politics. Whilst the public image of bioethics before the pandemic was that of liberal bioethics, the pandemic introduced features of public health ethics that were marginalized and not considered “bioethical” in the public eye.

This is most evident in relation to vaccinations. Vaccinations are among the most important public health achievements. Yet vaccines also raise a host of uniquely challenging ethical, legal, and policy-related questions. Like any medical intervention, vaccines carry the small risk of severe side effects sometimes. Unlike most other procedures, however, vaccination is performed on healthy individuals. Vaccination policy exemplifies tensions at the heart of public health in democratic societies and can indeed serve as a lens through which to explore central issues of public health ethics: the balance between the rights of the individual and the claims of the collective, the acceptability of compulsory measures, and the trade-off between risks and benefits in implementing a population-level intervention [32].

While resistance to vaccination and the tensions described have been present since the first introduction of smallpox vaccines, the COVID-19 era has highlighted the tension between liberal bioethics and new bioethical concepts and the tension between the ethical foundations of liberal bioethics and those of public health [31]. This tension was primarily expressed in an overly rapid and frenzied shift between the liberal principles of preserving human rights and privacy and the collectivist principles of public health and a social policy that included infringing on the right to freedom of movement, comprehensive lockdowns that led to economic crises, invading the privacy of isolated individuals, and more. Thus, considering the repeated waves of COVID-19 and the dual burden created by the pandemic coupled with seasonal illnesses, the normative guidelines became focused on how to “live with COVID-19”. More and more voices around the world called for adapting to the chronic pandemic as part of daily life [33,34,35]. Just as chronic diseases have replaced infectious diseases as the hallmark of sickness in the Western world, the chronic approach of “living with” is becoming the hallmark of public health in the era of the “new normal”.

In fact, the shift to a chronic approach to healthcare, i.e., living with an ongoing situation that is neither “healthy” nor “sick”, undermines the clinical approach to health and sickness. This is particularly significant when dealing with a chronic public health situation rather than an individual’s chronic health situation. First, there is a shift in the very concept of health. This perceptual deviation raises many questions—for example, what defines a healthy population? Is it one with a low number of sick individuals, or one that manages to cope with a chronic situation involving health risks that cannot be eradicated? In this context, the COVID-19 crisis has merely heralded a change in our approach to health and healthcare. Thus, for example, the implications of climate change can also be viewed as a chronic situation that requires us to change our definition of health and sickness and cope with it as part of the “living with” paradigm shift [36]. It is even possible that the implications of climate change, such as rising temperatures and the formation of new habitats for pathogens, will have a broader impact than that of the current crisis. Therefore, resilience should be addressed not only on the level of the individual but also on that of the community [37].

How are we to frame our approach to public health as we face these challenges and considering a chronic approach to healthcare? The shift in our understanding of health as a chronic situation lacking a clear distinction between sickness and health also brings new meaning to our concept of life—the “bios” at the foundation of the term “bioethics”. This gives rise to the question of whether public health and the collectivist approach on which it is based offer a practical and theoretical toolkit for handling this paradigm shift. How does public health contribute to the way liberal bioethics understands and copes with these types of chronic situations?

One possibility is that it expands the perception of bioethics beyond healthcare provision and the use of advanced medical technologies or even public health systems to encompass questions pertaining to the social factors shaping our concepts of sickness and health. These factors include not only the analysis of the economic, cultural, and even historical structure of health and sickness but also the politicization of these concepts. In the context of the COVID-19 pandemic, this expansion is expressed not only in the vaccination gaps within and between countries but also in the way we are going to heal our societies from the emotional, social, and economic burden of the pandemic. This in turn can be reflected to current heated debates such vaccine hesitancy and vaccine resistance. We propose a new bioethical approach to the question of vaccination that provides an alternative to the individual–society dichotomy and is centered on the concept of solidarity.

## 3. Vaccination Hesitancy and Gaps

In recent decades, more attention has been paid to the complex phenomenon of vaccine resistance, not being a monolithic reaction, and also to vaccine hesitancy, a much wider reaction. Those responsible for vaccination programs have adopted a wide variety of approaches to achieving high levels of coverage. These have included traditional health education and promotion campaigns in the mass media, recommendations given to patients by individual practitioners and issued by official medical societies, incentive programs that reimburse healthcare providers who achieve high coverage among their patients, installation of a network of mother and childcare centers with free delivery of vaccinations, and compulsory measures such as mandates for immunization prior to school entry. All these approaches attempt to strike a balance between the potentially competing values of respecting individual choice and assuring a sufficiently high degree of population immunity.

It is important to note that COVID-19 vaccines, like any other vaccine, are only part of the solution. Eradicating pandemics, or at least reducing their impact, depends on a broader aggregate of factors that are primarily geared toward strengthening the infrastructure of public health and public medicine systems, as well as environmental and social variables that affect health. The use of COVID-19 vaccines as a preventative solution, allowing for the possibility of going back to normal life once vaccination rates were high enough, caused concerns among many due to Pfizer and Moderna’s innovative mRNA-based vaccines. While these vaccines have since been granted emergency approval by the FDA and have undergone meticulous testing for safety and efficacy in many countries, there is still a lack of findings regarding their long-term effects, which presents an obstacle to building the public’s trust in their safety.

This has led to a series of wide vaccination gaps that have formed since the vaccines became available. The gaps between countries follow the lines separating the wealthy Northern and Western countries from the rest of the world. The race for vaccines was conducted by countries rather than by international bodies in what was referred to as “vaccine nationalism”, where each country used its power to secure vaccines exclusively for its citizens [38]. This situation created global vaccination gaps that ultimately led to the formation of mutations, new pandemic waves throughout 2021, the extended duration of the pandemic, and a rise in the number of victims [39]. Within countries, vaccination gaps stem from, among other things, unequal access to healthcare information and resources, including access to local clinics and vaccination centers. However, these gaps are also related to social, cultural, and political factors that raise doubts and concerns regarding the vaccines, leading to what is referred to as “vaccination hesitancy”.

Vaccination hesitancy—a middle category between those willing to be vaccinated and complete vaccination refusers—is a well-known phenomenon [40]. This middle category is of critical importance in every vaccination campaign, and certainly when dealing with a new vaccine. According to a work team organized by the World Health Organization in 2014, this vaccine-hesitant group needs be approached on three general levels—the contextual, the individual, and the group—as well as on the level of vaccination-specific influences [40]. Other studies suggest that the causes of vaccination hesitancy are difficult to pinpoint as they change according to the type of vaccine, location, population, and time [41,42].

Raising the level of trust in the safety of vaccines is an important step in coping with vaccination hesitancy. This trust is mainly related to the transparency of the process, straightforwardness regarding vaccination goals, accessibility of appropriate information appropriately for various populations, and comprehensive knowledge regarding side effects [43]. Regarding the COVID-19 vaccination for example, concerns about the new vaccine’s unknown long-term effects should be addressed. Worries about vaccines often stem from the dissemination of conspiracy theories, although this is a more significant factor for those who refuse the vaccine for ideological reasons than for those who are hesitant about them [44]. A preliminary study conducted in Israel even before the drug companies published their findings on the COVID-19 vaccines showed that the uncertain risk involved with vaccination was a crucial factor in the decision of whether to get vaccinated, also among medical teams [45]. As part of the effort to increase trust and transparency regarding the safety of the COVID-19 vaccines, leading public figures in different countries began getting vaccinated in December of 2020.

Trust is a fundamental condition for public responsiveness to policymakers’ decisions, both in the field of public health in general and regarding vaccination in particular [46]. Trust-promoting propaganda involves addressing the issues the public is concerned about rather than focusing on scientific issues that experts believe should be detailed and explained. The public’s vaccination hesitancy does not necessarily stem from a lack of information. Rather, people are often overwhelmed by a broad range of sometimes conflicting information coming from various sources. They also must cope with conflicts that change from one vaccine to the next. These include vaccine safety and efficacy versus the risks of the disease, one’s personal interest versus contribution to the community, and, in the case of COVID-19, one’s global contribution to ending the pandemic [47]. Concerns about the new vaccine need to be addressed through an explanatory campaign that does not ignore or necessarily dispel these doubts. Questions that cannot yet be answered also need to be addressed. For example, regarding COVID-19 vaccines, issues remain about their ability to prevent infection, the duration of immunity, and the extent and speed at which the virus mutates, beyond their efficacy in reducing morbidity and mortality [48].

However, beyond the question of trust, we would like to address the sociological aspect of vaccination hesitancy and suggest a direction that comes from the perspective of public health ethics. Vaccine coverage has sociological characteristics. Thus, for example, in Israel, the Orthodox population was found to be hesitant about childhood as well as seasonal vaccines [49], and the Arab population was found to have relatively high childhood vaccine coverage and lower seasonal vaccine coverage, often because of accessibility problems due to gaps in service availability [50]. Vaccination hesitancy was found to be more prevalent among the Jewish population compared to the Arab population, with “anti-vaxxers” often hailing from the well-established, educated strata of society who have a structural suspicion of conventional medicine. Moreover, the special structure of the health maintenance organizations in Israel, their deployment, and the central role they play in providing primary care make vaccines more accessible to all parts of society, including the periphery [51].

## 4. Moving Away from the Individual–Society Binary

Based on the studies cited above, it is clear that vaccination hesitancy is a social act no less than it is a personal one, meaning that it is expressed and even enhanced within the broader social context of the community. Therefore, we seek to move away from the individual–society dichotomy and argue that the vaccination hesitancy of any individual is also a result of their social standing and community belonging. Hence, the principle of autonomy does not stand alone as an individual’s independent way of thinking but stems from the norms regarding vaccinations that exist in each social surrounding, be it their community, family, congregation, or any other significant identity group.

If one of the sources of vaccination hesitancy is the identity group, one’s community, we propose that referring to this social structure and not only to the abstracted individual is essential in meeting the problem of hesitancy and trust in vaccination policy. However, we are all embedded in a pile of social structures, multiple belongings, and intersectionality. “Society” is an abstract and broad concept that is disconnected from one’s individual experience. In contrast, “community” is a more immediate concept that has clear and tangible boundaries that are missing from the concept of society and how it is perceived by the public. Communal thinking means thinking about giving and receiving on a more intimate level, where the individual generally gives and receives to members with whom they identify and experience a sense of belonging. In this way, the paradigm shift from the individual–society dilemma to a community-based understanding of vaccination policy brings the two ends together. Vaccination hesitancy is no longer an issue pertaining to the isolated individual on one hand or the broad population, society, or nation on the other.

The term “community” has many different meanings. “Communitarianism” is a well-established school of thought in the field of political science that holds that community is a collection of individuals with rights who unite around a common identity, interest, or purpose. Another concept of community stems from a more republican tradition that considers the community as the fundamental condition for social and political life, preceding the coalescence of individuals around a particular issue. In this sense, the community is more like a family where the relationship between members is based on camaraderie and solidarity.

In the context of vaccination policy, we prefer the latter meaning of community. The normative framework for alleviating vaccination concerns should be one of solidarity. It is important that we do not overlook or dismiss concerns or fears regarding vaccines, nor the important role trust plays in alleviating them. We believe solidarity, which centers on risking oneself for the public good, is more effective when that collective is an intimate community with which the individual deeply identifies.

In Israeli society, family is perceived to play such a central role that sociologists consider it a familial society [52]. In a society divided by religion, ethnicity, and national affiliation, the concept of family has been found to be a central value shared by most Israelis. In a study conducted on the 2013 explanatory campaign for polio vaccines (OPV), it was found that the Health Ministry sought to encourage vaccination by appealing to the value of family (its actual slogan was: “Just two drops and the family is protected from the risk of polio”) [53]. In this way, the Health Ministry sought to overcome the fact that those receiving the vaccine did not actually need it, the extra protection being intended to protect at-risk populations. Another study found that as part of the 2013 OPV campaign, pro-social motives played a central role in encouraging vaccination [54].

Accordingly, it is better, for example, to adopt the terminology of “community immunity” rather than “herd immunity”. The latter originated in the veterinary world of the early 20th century and was later used in relation to human beings. However, there are currently voices calling for this term to be replaced by one that emphasizes the community and population. In contrast to the term “herd”, which connotes a loss of identity and utilitarianism that can erase the individual, the focus on the individual’s place within their community creates a different frame of reference. Therefore, it is possible to design an explanatory strategy about vaccines that is adapted to the community (in terms of language, community leaders, and mainly protecting one’s neighbors and family). When a person understands that, beyond protecting themselves, vaccination protects those who are close to them, the dimension of mutual commitment that exists within families and communities increases. Vaccination itself should be presented as a communal act and a demonstration of solidarity. We believe framing vaccination as an act that demonstrates our fellowship as human beings who are vulnerable to diseases, while focusing on more intimate circles of closeness such as one’s neighbors, family, street, synagogue, etc., presents a window of opportunity for fostering vaccination solidarity. We hope vaccination will come to be understood as a communal act and that this will also create the social norm of getting vaccinated as the right thing to do on the level of the individual as well as the collective.

Health policy decisions that prioritize different populations in providing vaccines—decisions that may raise suspicion—can also take on a new meaning of social solidarity when understood in the context of the community, family, or neighborhood. In this sense, every vaccinated person knows the people who belong to at-risk populations; thus, vaccination becomes an act of empathy toward those populations done for their benefit. Similar to how the term “social distancing” was changed to “physical distancing”, as social distancing is undesirable in times of crisis, let us adopt the term “community immunity” rather than the old term of “herd immunity”.

## 5. Solidarity and Vaccines—A Bioethics Based on Positive Liberty

To inculcate the willingness to get vaccinated, the concept of solidarity must be emphasized as a central value of bioethics. As mentioned in the first part of this paper, this involves a shift from a liberal to a collectivist approach to bioethics. However, the concept of solidarity is multifaceted, and its definition sometimes shifts from a descriptive to a normative level, lacking the analytical dimension [55]. It should be clarified that a commitment to solidarity does not involve the individual nullifying and sacrificing themselves for the common good, an approach that views the individual and society as two mutually exclusive terms. Solidarity primarily refers to acting to benefit the social group with which the individual identifies and feels a sense of belonging [56,57]. Most solidarity researchers distinguish between different types of solidarity and different levels of solidarity acts. However, the common thread among these distinctions is that solidarity works on three levels: the interpersonal, the group, and the institutional [58,59,60,61,62]. The levels are differentiated according to the social institutions involved in mediating the act of solidarity. The interpersonal level involves providing informal help to someone we know, the group level involves acting to benefit others in our identity group, and the institutional level pertains to acts of solidarity involving a collective responsibility to the public, generally the residents of our country. Some researchers address solidarity with groups that are far from the civil boundaries of countries, in what is referred to as global or cosmopolitan solidarity [62].

Solidarity as a value of bioethics has limitations, the most salient of which is that to the extent that it promotes social inclusion, it also marks the boundaries of social exclusion. Ethical debates center on the boundaries of solidarity and whether it is beneficial to bioethical thinking [62]. Notwithstanding, we suggest that solidarity is of central importance to bioethics, especially considering the pandemic and global environmental crisis. From an analytical standpoint, employing the concept of solidarity emphasizes the individual’s positive rather than their negative liberty. If negative liberty means demarcating a space around the individual that protects them from the coercive power of society, positive liberty—as it is understood within the political philosophy of Rousseau and up to contemporary republicanism—is the individual’s ability to realize their capabilities, talents, and potential within the social framework [63]. According to this approach, solidarity is one of the central expressions through which the individual realizes their freedom to act for the benefit of others. Prainsack and Buyx [56,57] emphasize that solidarity is not just an expression of empathy but a practice that carries with it prices the individual pays for the benefit of another. A practice that involves a certain degree of sacrifice is the pragmatic definition of solidarity.

Solidarity presupposes similarities between the individual and the group they identify with and feel solidarity for. Therefore, it is highly important to cultivate this practice within communal identity frameworks that create a sense of familiarity and belonging. Regarding vaccination, cultivating a concept of solidarity as one that does not negate the individual but rather enables them to realize their sense of collective identity and positive liberty could mitigate the tension between the individual and society; hence the importance of using new wording, such as replacing the term “herd immunity” with “community immunity”, and adjusting and tailoring explanatory campaigns to localized terminology and sector-based cultural language.

## 6. Conclusions

Prainsack and Buyx define solidarity as a concrete move that one does for the sake of others. This move carries some sort of a “cost” or a “sacrifice” that charges the act with significant meaning. We identify this approach to solidarity with the abovementioned notion of “positive freedom”; an active participation in the making of the public good. However, both terms, “solidarity” and “freedom”, still carry with them the image of the individual–society binary. Society is a multiple phenomenon; there is no one society and not one public but rather publics, and accordingly, there cannot be one form of solidarity but rather many forms of solidarity that often intersect with each other, creating circles of inclusion and exclusion, in-groups and out-groups.

However, the times we live in, the times of the “new normal”, compel us to forge alliances to confront looming threats. Thus, for instance, the transition to community solidarity does not cancel the need for national civil solidarity, and in cases such as a global pandemic or coping with the climate crisis, there is also a need for cross-border solidarity. In fact, these may be complementary. For this to happen, the ethical foundation of community solidarity must be inclusive rather than exclusive. In other words, this solidarity must be based on a presumption of similarity and camaraderie with those who are close to us rather than delineating boundaries excluding those who are not part of our group. When facing crises, there is great potential for solidarity based on a broad foundation of human beings coming together for the purpose of coping with the randomness of natural disasters. Therefore, we need to make efforts to establish and better understand the role of solidarity as a central value in the bioethics of the “new normal”.

## Data Availability

Not Relevant.

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
