# Peer review of "Into the “New Normal”: The Ethical and Analytical Challenge Facing Public Health Post-COVID-19"

_ijerph, 2022, doi:10.3390/ijerph19148385_

Round 1
Author Response
Response to reviewer 1:
Thank you for sending me this interesting paper to read; it addresses an important issue. Generally, I think the authors have done a good job, but some changes are required before publication can be recommended.
- The authors do not acknowledge satisfactorily the complexity of bioethics. Even preCovid, there was much work on the social complexity of health in relation to bioethics; their work is not as novel as they suggest. The “new” bioethics they champion in the paper has been under development for many years prior to Covid-19 – this should be more directly acknowledgement. The most obvious example is public health ethics which the authors mention, but need to unpack more. Looking at the historical development of public health ethics would help. I have also done work on the relationship between patient centered care and citizen engagement and vaccination and the need for ethics to work across health systems, but there are other examples.
Answer: we agree with the reviewer on this point, and we revise our discussion on bioethics to acknowledge the collective aspects that exist in bioethics long before the pandemic. Notwithstanding, we still hold the view that the pandemic "pushed" these aspects to the forefront of bioethics and public health. For the revised discussion see changes in section 2 (towards a new bioethics).
- As part of introducing more on the historical development of public health ethics, suggested above - in terms of its response to communicable and chronic disease - acknowledging the roots of Covid “exceptionalism” in HIV “exceptionalism”. The authors mention the emergence of public health, but the main issue for them needs to be the development of public health ethics.
Answer: We added a detailed paragraphs on p. 3 to the history of public health ethics in the last two decades with an emphasis on vaccination policy.
- The above points should help the authors unpack their phrase “while not new”.
Answer: Although our additions and corrections to the paper, we still want to remain with this phrasing.
- Giving more importance to the role of ethical disagreements in vaccine hesitancy would help to highlight the importance of values-based debate on this issue: people objecting to state interventions and limits of freedom.
Answer: Please see our detailed discussion on vaccinations and public health in relation to interventions and restrictions on p.4 and the section on vaccine hesitancy (p 5-7).
- Given the focus on solidarity, discussion of autonomy could be usually supplement by discussing relational autonomy … another example of ethics being more complex that the paper acknowledges.
Answer: We added reference to relational autonomy and discussed other collective elements in bioethics, see section 2.
- The authors recommend turning to solidarity as a central principle in the “new” bioethics. Solidarity is already a key principle in public health ethics. Yet during the Covid pandemic the existence of public health ethics has not really helped to manage the liberal, rights-based protests against public health measures. This problem needs to be acknowledged, there is a profound deep-rooted conflict at the heart of liberal democracies will not be easily overcome.
Answer: Thank you for this important comment. We acknowledge the shortcomings of public health efforts to reach wide compliance. In the conclusion section we call for a more reflexive and sensitive approach to solidarity which does not account for society as one whole but rather to publics and solidarities. We hope that this form of solidarity would yield more trust in public health policies.

Reviewer 2 Report
Dear authors,
Thank you for the opportunity to review this excellent paper, and indeed as argued by various papers, we need push forward the political analysis of health, recognising the important role of political, social, and structural determinants of health – beyond the individual choice/responsibility. This paper offers a new perspective to discuss the public health solidarity and new approaches to bioethics during/post COVID-19. I have several comments and questions that hopefully will improve the quality of this paper.
· In the abstract – the argument is justifiable; however, the authors need to provide sufficient information as to why ‘the new normal’ may require a new lens/system of bioethics? Furthermore, it is not clear in the abstract regarding how solidarity can be strengthen in the contexts of new normal?
· In the introduction: (1) I would like to see in the introduction how the conception of health evolves throughout the pandemic (e.g. before and after the pandemic), and explain why these various conceptions/approaches to health exist in the first place. This will set a scene for your next sections. (2) What are the consequences of these various approaches to health both for public health interventions/programs/policies and health outcomes of the population?
· As the centre argument here is COVID and post-COVID, I strongly recommend the authors to explore how COVID has changed the way our health systems operate and what is the state of relationships between human/individual and society related to health matters before COVID and how COVID influences this relationship.
· Various conceptions of new normal exist, perhaps it is a good idea to define new normal in this context - new normal from whose perspective are we talking about in this context?
· Solidarity - how would you define solidarity, and the exercise of power from global north countries to global south countries will influence such public health solidarity?
· What is the state of PH solidarity before COVID? How COVID has changed the definition and conception around public health solidarity?
· Are there any tactical and strategic recommendation to strengthen PH solidarity moving forward?
Author Response
Response to reviewer 2:
Comment 1: In the abstract – the argument is justifiable; however, the authors need to provide sufficient information as to why ‘the new normal’ may require a new lens/system of bioethics? Furthermore, it is not clear in the abstract regarding how solidarity can be strengthen in the contexts of new normal?
Answer: Please see the revised abstract. We added two sentences, explaining our view on the necessity of solidarity as a normative concept in "the new normal".
Comment 2: In the introduction: (1) I would like to see in the introduction how the conception of health evolves throughout the pandemic (e.g. before and after the pandemic), and explain why these various conceptions/approaches to health exist in the first place. This will set a scene for your next sections. (2) What are the consequences of these various approaches to health both for public health interventions/programs/policies and health outcomes of the population?
Answer: Thank you for this comment. We added a description of the evolution of public health interventions during the pandemic and the way the concept of health changed during the pandemic. See upper paragraph in p.2. We argued there that while the measures taken in confronting the pandemic were not new and included quarantines, lockdowns, and a series of restrictions, they encountered suspicion and caused controversies and disputes and destabilized the trust in concept of health. See also the quote on p.2 from one of NEJM editorial members on how the pandemic changed the way he understands health. In p. 4-5 we added a more detailed discussion on vaccination policies during the pandemic, its sources, and difficulties. We thought that discussing all these aspects in public health and the concept of health in the introduction would be slow down the text's flow.
Comment 3: As the centre argument here is COVID and post-COVID, I strongly recommend the authors to explore how COVID has changed the way our health systems operate and what is the state of relationships between human/individual and society related to health matters before COVID and how COVID influences this relationship.
Answer: Although we did not scrutinized the actual shifts and modifications that medical fields and disciplines have undertaken during and after the pandemic, the paper points to what we see as a significant shift in bioethics that took place during the pandemic and we thought is of major importance and interest to the journal's readership. We described the ethical modifications that are necessary for public health in the era of "the new normal".
Comment 4: Various conceptions of new normal exist, perhaps it is a good idea to define new normal in this context - new normal from whose perspective are we talking about in this context?
Answer: We add a definition of the "new normal" on p.2 second paragraph.
Comment 5: Solidarity - how would you define solidarity, and the exercise of power from global north countries to global south countries will influence such public health solidarity?
Round 2
Author Response
Thank you for reading again carefully our manuscript and commenting on it. We learned from your comments and revised the manuscript, and we hope that now it meets your reservations. You urged us to be "more nuanced and sophisticated" in our criticism on bioethics and to acknowledge its complexity. Specifically, you addressed our efforts to unpack "yet while not new" claim. Accordingly, we modified sections in the paper that relate to bioethics complexity and moderated our claim about the newness of public health ethics. Our argument is not that public health ethics is a new phenomenon that resulted from the pandemic. We are fully aware to the critics on bioethics that were raised long before the pandemic (and even wrote some as well). Our claim is that the covid-19 mainstreamed public health ethics to be more present, more visible, and more relevant in discursive fields and spheres of actions that usually viewed bioethics as concentrated in individualistic perspectives. The mainstreaming of public health ethics is expressed in the call of "health in all policies" that was introduced during the pandemic as well as in the debates on public health outside professional and academic circles. The pandemic brought public health ethics to the front of public discourse. This change in the position of public health ethics in the public sphere is one of "the new normal" features and we call for even more integration of public health ethics. Furthermore, we call for a further development of these ethics to include references to publics (rather than public) and to different forms of solidarities.
Specifically, we changed the phrase on p. 2 and modified the introduction. Our changes are mainly concentrated in the introduction and section 2 (especially second paragraph on p.3 after "towards new bioethics" and top and bottom paragraphs on p. 5). In fact, we ask you to focus mainly on pages 1 – 6 where we inserted most of our changes.
Thank you for investing time and effort in reviewing our paper.
The authors